# Complementary Currencies for Humanitarian Aid

Leanne Ussher [1,2,3,*], Laura Ebert [4], Georgina M. Gómez [5] and William O. Ruddick [6,7]

1 Department of Management, Politics and Philosophy, Copenhagen Business School,
2000 Frederiksberg, Denmark
2 Wolfram Blockchain Labs, Wolfram Research, Champaign, IL 61820, USA
3 Center for Civic Engagement, Bard College, Annandale-on-Hudson, NY 12504, USA
4 Department of Economics, State University of New York at New Paltz, New Paltz, NY 12561, USA;
ebertlc@newpaltz.edu
5 Institute of Social Studies, Erasmus University Rotterdam, 3062 Rotterdam, The Netherlands; gomez@iss.nl
6 Grassroots Economics Foundation, Kilifi 80108, Kenya; willruddick@gmail.com
7 Institute for Leadership and Sustainability, University of Cumbria, London E14 6JE, UK
* Correspondence: lussher@bard.edu

**Abstract:** The humanitarian sector has gone through a major shift toward injection of cash into vulnerable communities as its core modality. On this trajectory toward direct currency injection, something new has happened: namely the empowerment of communities to create their own local currencies, a tool known as Complementary Currency systems. This study mobilizes the concepts of endogenous regional development, import substitution and local market linkages as elaborated by Albert Hirschman and Jane Jacobs, to analyze the impact of a group of Complementary Currencies instituted by Grassroots Economics Foundation and the Red Cross in Kenya. The paper discusses humanitarian Cash and Voucher Assistance programs and compares them to a Complementary Currency system using Grassroots Economics as a case study. Transaction histories recorded on a blockchain and network visualizations show the ability of these Complementary Currencies to create diverse production capacity, dense local supply chains, and data for measuring the impact of humanitarian currency transfers. Since Complementary Currency systems prioritize both cooperation and localization, the paper argues that Complementary Currencies should become one of the tools in the Cash and Voucher Assistance toolbox.

**Keywords:** Complementary Currency; humanitarian aid; cash transfers; Red Cross; backward linkages; blockchain; cryptocurrency; Kenya

## 1. Introduction

It was time to try something new in delivering humanitarian and development aid. That is, modalities that build resilience in the long term, promote collective action among aid recipient communities, and increase effectiveness by using new technologies. That was the intention when the Red Cross partnered with Grassroots Economics (GE),[1] a non-profit foundation operational for over 10 years in Kenya, and implemented a Complementary Currency (CC) system. CC systems are networks of producers and traders that exchange goods and services with each other using self-organized means of payment of voluntary acceptance (Blanc 2012; Gómez 2019). CC systems have various aims, such as reconnecting communities by exchanging favors, rewarding socially desirable behavior (for example, volunteering, recycling, care of public areas) and promoting local development and productive capacity a lá Hirschman (1958). CC systems have been part of the monetary ecosystem for several decades and they are known to empower communities and increase resilience (Huber and Martignoni 2013; Scott Cato and Suárez 2012), but they rarely appeal to humanitarian organizations to deliver aid because they are not widely known or well understood by traditional economists and policy makers (Fare and Ahmed 2014). This research focuses on a Complementary Currency (CC) in Kenya issued by GE,

which since 2010 has been innovating CC systems in coordination with development aid donors. GE is at the cutting edge of using new digital technologies on a public blockchain for developmental aid and offers a rich database for research in this area.

The introduction of new modalities of aid delivery addresses long-standing concerns among international organizations, such as the ways of targeting aid to recipients who need it the most and maximizing development impact and cost effectiveness for each dollar spent. These concerns relate to receiving funding from different donors with diverse goals and priorities while designing and developing aid programs tied to the particularities of the targeted communities, planning proper logistics at a grand scale, getting the necessary infrastructure and labor to implement them, delivering the most appropriate aid modality to the beneficiaries in crisis areas and finally moving beyond aid to sustainable development (Holguín-Veras et al. 2012a, 2012b; Michel 2016; Ward and Lewis 2002; Zambrano et al. 2018). Despite their thoroughness, aid programs have not always reached their targeted populations and when they do, provision of essential food and development aid has been known to cause a myriad of negative side effects such as undermining production by the local economy, affecting consumption patterns, distorting prices (Kripke 2005), or fostering dependency systems (Kabonga 2017; Moyo 2010). Moreover, a critical problem is how to give beneficiaries agency to build resilience and stand on their feet once the aid concludes (Pelling and High 2005), and more important, how aid can be leveraged for the whole community with a focus on the positive externalities or social impact of aid and its measurement over the long term.

These challenges have triggered the experimentation and testing with new technologies, including innovative software, big data, artificial intelligence, blockchain, and other technological solutions. In recent years, mobile and distributed ledger technologies have enticed multilateral organizations to explore financial developments with bespoke digital protocols that reduce program costs and support self-sovereign data (Allende López 2020). For example, UN agencies such as UNWomen and WFP[2] are working on projects with biometrics, using blockchain-based solutions to deliver aid to the Syrian refugees in Azraq and Zataari camps of the UN. UNICEF[3] has released a technological solution based on blockchain to receive, manage, and distribute donations of Ether and Bitcoin cryptocurrencies. UNICEF[4] has used machine learning and artificial intelligence to maximize childhood vaccination and predict supply chain needs. Within that trend, blockchain based CCs are being piloted by the Red Cross societies and Grassroots Economics Foundation in Kenya.

In the landscape of humanitarian and development assistance, there has been a dramatic move towards the provision of cash, or vouchers exchangeable for goods and services, directly to aid recipients. The Cash Learning Partnership (Jodar et al. 2020) states that cash and voucher assistance (CVA) was US\$5.6 billion in 2019 and constituted almost 17.9% of total international humanitarian assistance (IHA). This was double what it was in 2018—US\$2.8 billion and 10.6% of IHA. The move by donors towards CVAs has been driven by cost efficiency, effectiveness and accountability, as well as a greater appreciation of recipient perspectives and their autonomy. In this paper we argue for a local CC to be recognized as a financial aid modality that promotes social collaboration and commitment towards localization and resiliency. To date CCs are missing from the literature, but they offer a framework that builds upon the cash transfer system.

This study started with a desk research of documents and evaluations, in addition to academic papers on CCs, the particular experiment of GE CCs, and its antecedents. In a second instance, blockchain-recorded CC user transaction data from GE, which is open source, was cleaned and studied using Wolfram Mathematica, visualizing supply chains, and mapping their development overtime from late 2018 to mid-2021. Attention was paid to the real side activity of trade networks in producers' sector and regional locations. The product-space was used as a heuristic for understanding second, third and higher order effects from aid in the form of CCs as they circulate and grow a community's local economy from within. Finally, a discussion was held with the Grassroots Economics CC program directors on the history of their program since 2010, and the interpretation of our findings.

The study shows the potential for deploying a novel and new humanitarian cash modality in the form of CCs. This paper is authored by three scholars and the founder of Grassroots Economics Foundation.

This paper begins, in Section 2, with a comparison of the functions and impacts of unconditional cash versus conditional cash or voucher assistance. We then compare these two CVA modalities to a third modality called CCs, which can leverage aid dollars by reducing financial leakages and increasing local income, local employment, and coordinate local decision making or 'commoning' (Meyer and Hudon 2017). In this section we also simplify CC architecture into two types—mutual credits versus convertible currencies. The third section of the paper summarizes a theoretical aspect of economic development that should be central to developmental aid programs—the importance of forward and backward linkages in creating 'productive knowledge' that stimulates local economies and reduces aid dependency. Albert Hirschman (1958) in his strategies for unbalanced economic growth and more recently Hausmann and Hidalgo (2010, 2011) describe product diversity or 'product-space' metrics that are better predictors of economic development than measures of competitiveness, governance, education, and financial depth. Their complex systems growth strategies have many parallels with celebrated grassroots community organizer and local currency advocate Jane Jacob's own theories for economic expansion through a rich social fabric of local engagement and pathways. These theories provide the context in which we assess the progress of GE Kenyan CCs in Section 4. GE CCs have had growing traction and success in informal urban and rural slums of southern Kenya. Since January 2020 GE partnered with the Red Cross and expanded developmental aid in cash, vouchers, and CCs. Using open source blockchain data available since the CC went digital in September 2018, we document the dynamic evolution of this ecosystem to date. Using CC member business identity as a proxy for the types of goods sold, we show a growing diversity across the 'product-space' within the CC system, even during periods of declining participation. While further study is needed, our metrics suggest that CCs ideally create local forward and backward linkages and 'productive knowledge' or capabilities that stimulate local economies and reduce aid dependency. When re-localizing trade and production, aid can have a long-term impact that increases economic progress and local resiliency by reducing dependency on resources from outside the region—be that external goods and services, external finance, or foreign aid.

The World Humanitarian Summit, the Agenda for Humanity, and the launch of the *Grand Bargain* in 2016 (IASC 2021) were major initiatives to reform the humanitarian system and implement localization commitments that have existed for the past 25 years. This paper describes a pathway, and a case study, for aid independence and local economic growth with local CCs. We conclude that the specific goals of 'quality and localization' of humanitarian funding by the Grand Bargain should look deeper into new solutions such as CC systems.

## 2. Why Try Something New?

According to the United Nations Office for the Coordination of Humanitarian Affairs (United Nations OCHA 2019), humanitarian aid seeks to mitigate the impact of several crises. From 2014 to 2017 conflicts have increased the displaced population from 59.5 million to 68.5 million, natural disasters have been exacerbated by climate change and have affected 350 million people (UN Office for Disaster Risk Reduction 2020), while 120 million cannot secure minimal food requirements. Despite the increases in aid, it is estimated that 132 million people in 42 countries around the world will be exposed to crisis and risk situations. The IMF World Economic Outlook describes the contraction of economic activity in 2020 due to the COVID-19 pandemic as "unprecedented in living memory" (International Monetary Fund 2021). The economic devastation caused by the pandemic and containment policies show an output contraction of over 10% in several countries (International Monetary Fund 2021), with loss of income, affecting women, youth, and informal workers the most. The report estimated that an additional 95 million people have

fallen into extreme poverty in 2020. While governments hope that economic output and employment will bounce back, many have implemented whatever expansionary policies their budgets and political settings allow to address a widening and persistent gap between rich and poor. With a growing emphasis on resilient and sustainable systems, many are looking to civil society to fill the holes of government and private sector services or responsibilities (Ingram 2020). CC initiatives are among such responses to the pandemic at the local level, but overall remain quite small (Adriano 2021).

The support of grassroots initiatives and emphasis on greater responsibility and decision making by recipients themselves have followed the growing trend of the last decade towards direct cash transfers in national currency to aid recipients (unconditional grants in cash), distinct from non-circulating vouchers (defined as conditional cash grants), and in-kind aid. The first two categories fall under the umbrella of cash and voucher assistance (CVA).

World Vision International in 2019 reported that cash and voucher-based programs were the most popular modality of transfers by donors and international organizations and the most convenient. The crisis caused by the COVID-19 pandemic and its mitigation policies in the last year have increased the preference for cash transfers. According to a World Bank report (Gentilini et al. 2020), there were 98 cash transfer programs in the world in 2019, and in 2021 the number has increased to 277 in 131 countries. Unlike in-kind transfers of imported goods that compete with local suppliers and destabilize markets, or the provision of sanitation, health, and education services that bring experts and labor from outside, cash promotes democratic grassroots organizing that can mobilize local resources. With cash individuals have flexibility and the freedom to choose local products and build up local markets that can kickstart local supply chains.

Unconditional cash transfers are seen as the modality that offers the most freedom to beneficiaries, (Cunha et al. 2019; Gentilini 2016). Advocates argue that they are more efficient, more transparent, and more accountable than any other modality: "It gives the dignity of decision-making back to the people who know best what they need and kickstarts local markets and supply chains, helping economies get moving again". (Center for Global Development 2015). Despite this emphasis on the second order effects of aid—kickstarting local markets and supply chains—this paper will argue that at the present time, CC systems are superior in creating and measuring local expenditure multipliers, which allows for an effective demand management at the micro scale, with a narrow focus on special groups or projects.

### 2.1. Unconditional Cash Transfers

A Cash Transfer is defined by *The Cash Learning Partnership* (CaLP) as

" . . . the provision of assistance in the form of money—either physical currency or e-cash—to recipients (individuals, households or communities). Cash transfers are by definition unrestricted in terms of use and distinct from restricted modalities including vouchers and in-kind assistance". CaLP (2020)

The increase in cash transfer programs for aid delivery seems to have eclipsed other delivery modalities (Davis and Handa 2015) but cash is not unproblematic. It requires financial infrastructures that are not always available or cost a significant proportion of the funds that could help more people in need (Handa and Davis 2006; Holguín-Veras et al. 2012b). In order to secure the transactions and to manage the risks of handling cash internationally, aid organizations usually remit aid money through banks and financial intermediaries and then distribute this to the beneficiaries via local bank accounts. Such transfers take considerable time due to regulations on money laundering and risk management of financing terrorism for each of their clients. These financial services also extract funds that could otherwise benefit aid recipients; a WFP blog on the Building Blocks project refers to "enormous" banking costs (Dhameja 2019). Moreover, the use of cash raises risks of financial mismanagement, and some organizations recommend measures to minimize those risks. For example, Oxfam reports that physical cash transfer programs are at risk of

theft, looting, diversion by local elites, attention by authorities and warring parties, and personal attacks on the way home (Creti and Jaspars 2006). Oxfam hence transfers funds to beneficiaries' bank accounts directly or using mobile phones, limiting payment information to a few people, decentralizing distribution so that cash transportation would be unnoticed, or moving only small amounts of money with varied payment days.

Around the aid-cash nexus various new fintech companies and digital innovations are reducing costs, adding liquidity, building coverage, and reducing risks. Non-bank money transmitters or mobile phone companies and their agents can act as intermediaries for the 'unbanked, 'where payments are done through SMS services on mobile phones. Beneficiaries that receive cash transfers via mobile phones or e-wallet balances can withdraw, deposit, and transfer money without having a contractual relationship with a bank. When paper cash is required, the recipient would visit a retail agent to convert the electronic money into cash (Mas and Morawczynski 2009). The World Bank Development Report (World Bank 2016) confirmed that where the infrastructure is already set up (phone ownership, good network coverage, registered mobile wallets for KYC criteria, cash withdrawal infrastructure or high agent density) then mobile phone payments allow more secure and lower cost in their distribution, reduce cost of financial transactions, and potentially benefit the government because of digital records they can trace and tax. However, transaction fees on using mobile money, especially for low value transactions, has been a concern. In a 200-person survey among general users, 56% and 57% of respondents in Ghana and Uganda still prefer using cash over mobile money, mostly due to fees (Tay 2020). Fees are significant for the poor where small value transactions face up to 10% in fees. Fortunately, following the COVID-19 pandemic, tariffs on low value transaction bands (less than US$10) became temporarily free on mobile money. However, the risk remains that since traditional third-party financial services are not designed for, or dedicated to low income and undocumented customers, the poor will be forced to seek out predatory lenders or services as an alternative. Aid in the form of cash transfers can indirectly create a revenue stream for these third parties.

In a nutshell, cash transfers allow individuals to respond to their unique complex needs, can stimulate effective demand and offer efficient ways to reach people in need faster, and at lower cost, than other forms of emergency assistance. However, such transfers rely upon an accessible and inclusive financial infrastructure. The process presents various privacy and security risks that have the promise to be addressed by new technological solutions. Fintech corporations have promoted digital cash transfers in addressing the traditional aid modality challenges: targeting, verification, delivery, and monitoring of recipients. Widespread COVID-19 cash-transfer payments have raised the use and acceptance of unconditional cash transfers.

### 2.2. Vouchers or Conditional Cash Transfers

A voucher is defined by CaLP (2020) as

" . . . a paper, token or e-voucher that can be exchanged for a set quantity or value of goods or services, denominated either as a cash value (e.g., $15) or predetermined commodities (e.g., 5 kg maize) or specific services (e.g., milling of 5 kg of maize), or a combination of value and commodities. . . . They are redeemable with preselected vendors or in 'fairs' created by the implementing agency".

Vouchers often have an expiry date and are non-transferable. They are restricted by default, although the degree of restriction will vary on the basis of program design and type of voucher. While unconditional cash transfers are thought to be fairer, more cost effective, and more democratic than conditional cash transfers, the latter are implemented with the premise of building human capital (health or education) that can then produce a stream of returns in the future, thus helping the next generation, not just fighting current poverty. For example, Sarah Baird and coauthors (Baird et al. 2019) looked at vouchers for education in Ghana and found that school enrollment among families that got conditional grants rose by 41% on average in the various programs; the increase among those that got

unconditional grants was only 23%. If conditions were implicit or soft (e.g., if recipients were simply encouraged to take children to school), enrollment merely rose by 25%. If conditions were tough (e.g., if school attendance was mandatory), enrollment was boosted by 60%, "a big bang for the relatively few bucks involved" (The Economist 2018).

Vouchers generally require a third-party financial service company such as a bank or a non-bank like Sodexo.[5] Vouchers can also avoid rentier third parties through the use of blockchains. Hunink (2019) compares e-vouchers using a banking system versus using blockchain technology in the Building Blocks' project with Syrian refugees and notes that new technologies have solved several problems of delivering cash directly (WFP Innovation Accelerator 2019). If vouchers are used on a blockchain, third parties can be removed entirely and delivery can be cheaper, in theory.

While the critique of the effectiveness of vouchers often focuses on supply side issues (Are goods locally available? Are enough suppliers willing to contract with aid agencies to meet recipient demand? (Cunha et al. 2019)), vouchers can also be demand driven. Family planning and reproductive health or education voucher programs in Uganda and Kenya can be used at either public or private providers, promoting market-based outcomes. "When voucher clients bring in revenue, private sector providers are very interested and willing to deliver" (Arur et al. 2009). The power of vouchers, as with in-kind aid, is that technocrats can more easily match supply side availability with demand. For example, food vouchers have been tailored to support local farmers, and in Uganda Uber clients have received vouchers to spend with street vendors who were particularly vulnerable during COVID-19.

Hence, in contrast to cash, conditional cash transfers or e-voucher systems have both first and second order targets: recipients of the voucher and the supplier of goods and services who accepts the voucher. This creates a measurable and traceable link between demand and supply. While vouchers have less flexibility than universally accepted cash, the benefit is that they can direct and localize spending (if that is a goal) guaranteeing not just one but two income earners. However, the next transaction in the chain is unknown as there is no way of tracking how providers that get reimbursed spend their cash (Cunha et al. 2019). Whether the modality of CVA is the unconditional or conditional cash transfer, there is still the important problem of national currencies being prone to quickly leave communities (Santos 1979; Thome et al. 2016). The expenditure multiplier, or the number of times in which cash circulates in a community, has a central role for long term development aid. Indirect measurements of aid over the long run hint that cash injections (CVAs) generate modest long-term impact or none at all (Clemens et al. 2004).

*2.3. Complementary Currency Systems*

The "something new" advocated for in this paper is for aid entering a less developed region to utilize, build upon, and leverage local resources with a local monetary system, in parallel with the national currency. The implementation and longevity of CC systems are well documented. CCs are normally a grassroots' innovation restricted to the voluntary participants in a dedicated network of means of payment. It includes the *moneta sociale* in Italy, local exchange and trading systems (LETS), time banks in English-speaking countries, the *monnaies parallèles* or *Solidaire Economie Local* (SEL) in France, and the *Red de Trueque* (RT) in Argentina (Gómez and Helmsing 2008; North 2007).

A local CC or token is a form of 'voucher' or credit obligation redeemable for goods and services but one that is fungible, transferable and potentially accepted by all members of the CC community. Unlike an aid voucher, CCs can be spent and re-spent across all member businesses that accept them. An aid voucher connected to a local business is immediately cashed out after its use. While this injected national money might circulate, it might also be spent on goods or services from outside the community (we shall refer to these as external inputs or imports) and depart the local economy. A CC directly addresses the problem of leakage of national money from a local, less developed community. The purpose of CC aid is for recipients to spend them locally and raise the likelihood of earning

CCs back when the community's purchasing power folds back in on itself, creating circular chains of spending and income. CC systems can insulate local networks of producers and traders that exchange goods and services with each other when members continue to accept and spend this complementary means of payment, which restricts their purchases to local goods and services (Blanc 2012; Gómez 2019; Ryan-Collins 2011). Since a CC circulates in parallel to, rather than as a replacement for, the national currency, members may ultimately have more national currency as they limit expenditures of it on external inputs.

Broadly, CVAs and CCs are all forms of money. Amato and Fantacci (2012) describe money as having two opposing roles—liquidity or clearing. In the former money is a storable financial asset, in the latter money operates as a counting device to facilitate the trading of goods and services for the goal of reciprocity (everyone spends what they earn and aggregate spending equals aggregate earnings). Most CCs are designed for the purpose of clearing. They have use value in spending them, but have limited intrinsic value. CCs are typically stable in value and have no interest rates on saving or borrowing, as opposed to a national currency which has a liquidity preference or most crypto currencies created for speculative financial gain. However, some CCs are more liquid than others, that is, more easily convertible into a national currency which can divert their use as a medium for exchange. For example, Bristol Pounds as paper CCs were issued by local banks in exchange for national currency and supposedly 100 percent of this was 'available' for users to convert back at a 1:1 rate at any time. A similar but less liquid CC are Berkshares which if redeemed at a bank incur a 5% discount or fee which incentives spending them. They can be bought from banks at a 5% discount and this incentives their introduction into the ecosystem.[6] This spread represents a transfer of wealth from businesses who redeem them, to consumers who purchase them. Critics of liquid CCs argue that their ease of liquidation ('cashing out') reduces their local circulation and thus local income generation. Advocates argue that by having a guaranteed financial backing their acceptance is more widespread.

CCs that shun liquidity or withdrawal from circulation through conversion into the national currency are barter circles or mutual credits[7], which operate independently of collateral, and advocated by Greco (1990) based on Riegel (1944). Under this system money is created in the act of trade. In most cases members start at a zero balance that goes down by spending and up by earning—all credit and debit positions of the group at all times add up to zero. The 'money supply' of CCs is the sum of all debt positions—created through spending and destroyed through reciprocity (closing out debt and thus credit positions). Amato and Fantacci (2012) and Lucarelli and Gobbi (2016) describe these CCs as credit clearing systems, or *clearing CCs*, and their 'long run' equilibrium is the return of all member balances back to their starting point. The CC circulates in the community at par with the national currency, where the stability of its price depends on voluntary action by those who accept the CC for final payment at par. Its price is not guaranteed at a bank 'conversion window' as in the Bristol Pound or Berkshare case. Essential to their management, but often missing, are mechanisms to help members return back to their starting point in the 'long run' (Sardex is a good example of a mutual credit that dedicates resources to creditor and debtor matching services)[8]. In mutual credits it is common (but often mistaken) to assume that members are independent from each other and symmetrically distributed in needs and offers. Allowing withdrawal from a *voluntary* mutual credit often infers that an individual member who is a debtor that *cannot* earn CCs to return their balance to their starting point is asked to pay in national currency back to the platform, while a creditor rather than receive their 'excess positive balance' as national currency is often told to spend their 'excess positive balance' in CCs. This asymmetry is created to avoid the moral hazard of participants not circulating the currency and thus not mutually supporting each other in CC trading. Unlike convertible CCs, which have a trusted third party and collateral to easily resort to, mutual credits are based on common trust and prone to default risk (debtors leaving without paying back) and imbalances. Many mutual credit advocates (Slater and Jenkin 2016; Linton and Soutar 1994; Greco 2018) rely on a stable 'long run' equilibrium assumption, i.e., where individual credits and

debits, or excess imbalances, revert back to their initial starting point (typically zero). In reality this rarely exists and what primitive debit and credit matching tools exist, are often ineffective.

The build-up of imbalances is most commonly discussed in terms of a country's international capital accounts—if exports are greater than imports then a country has a surplus, while the opposite means a deficit and rising foreign debt. A high marginal propensity to import and/or insufficient export income to pay for imports is a dilemma that many countries face—defined as balance of payment constrained growth (Thirlwall 2012). A Keynesian macro interpretation will often blame surplus countries and their lack of spending for the deficit country constraints. If the surplus countries imported more from deficit countries, spending down their surplus, then the deficit countries would have the income to pay for their purchases and not go into debt allowing the system to converge back to equilibrium. But there is no reason for surplus countries to spend on deficit countries; they can instead spend with other surplus countries. Dependency theorists (Ghosh 2019) will also emphasize the drain of surplus or capital in a one-way direction from the periphery to the center. What is common is for a global north and south or east and west to become more divided due to virtuous or vicious cycles (positive feedbacks where rich get richer and poor get poorer).

At the regional or local level, especially evident in the periphery, vendors that depend on external joint inputs for their local outputs cannot easily substitute local goods when they do not exist. These members will see growing CC surpluses (excess positive balances). In addition, not all members that spend CCs before they earn, can earn the money back if surplus units are not spending with them. These net buyers may go into negative excess balances even further, in good faith that at some time in the future they will earn the CCs back, but in the end they drop out (default on their promise of reciprocity). More members joining and spending might mean a growing debt, forcing growing CC surpluses on active entities who do sell. The lack of common knowledge in a complex system means that even if an equilibrium exists it may not be stable—convergence is not guaranteed—and law suits to bring debtor members into accordance are expensive. When all balances are interdependent solutions like matchmaking or the filtering of membership to ensure matching partners is a never-ending process. Some will argue that Keynes' had the solution (proposed in his 1941 international clearing union) by making symmetric credit and debit limits and penalties. In practice, mutual credit CC systems are rarely symmetric, but even if they were, Keynes' solution did not guarantee convergence either. Ussher et al. (2018) argue that Keynes' original proposal was tied to commodity buffer stocks, and it was this that buffered imbalances and provided the counter-cyclical external effective demand to bring the system back to equilibrium.

Solutions for countering imbalances are likely to require some move along the spectrum towards the liquidity model which entails conversion options for some traders and withdrawal of CCs from the system. These options require funding from outside the system which is why matching CC systems with external aid is a good partnership model. Excess balances convertible into a voucher that is redeemable into some socially necessary goods at an authorized enterprise might be the best answer. For example, a mutual credit platform could partner with a public goods provider like a grain mill, a municipality, or a digital utility company, that would accept CCs in exchange for milling services, tax subsidies, or advertising, respectively. These goods or services that are demanded by everyone can resolve imbalances, if these authorized enterprises can redeem their vouchers for national money with the mutual credit platform (rather than recirculate CCs). Companies that need more customers to utilize their spare capacity, or have room to raise taxes/fees, or have decreasing marginal costs will have an interest in negotiated partnerships with a mutual credit platform with a large local membership.[9]

### 3. Backward and Forward Production Linkages

Poor communities with low investment, low adoption of technology, and a lack of entrepreneurial innovation will naturally have a lack of money coming in through investment or exports. Low production and the need to import commodities from outside the community can easily lead to a dependence on external production. When money flows out faster than it flows in, there is a poverty, commodity, and balance of payments trap. High levels of debt will mean a growing humanitarian crisis and/or a high level of aid dependence. CVAs in such systems as cash or second order cash (vouchers) will quickly leave the community, flowing to the trade and financial centers, and to profitable businesses in locations where dense networks leverage investment and are more productive. Along with the periphery being drained of funds, human and physical capital will also tend to drain out, partnering with external technology for higher productivity pathways.

These financial leakages characterize the national accounts of the Global South but can also exist within a country. For example, in the aftermath of the American Civil War the scarcity of credit and currency during the harvest in rural areas drove prices down and caused debt-deflation spirals (Mehrling 2011, p. 31). Similarly, urban informal-sector slums and ghettos where public goods are scarce, wages are low, and unemployment is high face a glut of resources for the amount of money available and an incapacity to coordinate a way to utilize these resources effectively. While in need of humanitarian assistance, we argue that it is unlikely CVAs, denominated in a national currency, will break the development trap of these communities.

Regional economies get stuck in poverty traps not just because they do not have money, but because they do not have 'money that stays local'. Ward and Lewis (2002) at the New Economics Foundation called this the "leaky bucket dilemma" where cash injections leak out: savings are redirected to external investment; spending is directed to goods from outside the community; skilled workers from outside the community are employed instead of local skilled labor; and the latter have an incentive to migrate out.

In a similar vein, Albert Hirschman argued that lack of economic progress is typically not because of a lack of resources, but rather on the lack of a "binding agent" that can call forth and enlist for development purposes resources and abilities that are "hidden, scattered, or badly utilized" (Hirschman 1958, p. 5). This paper presents CCs as accomplishing both goals—they are 'money that stays local' and they follow Hirschman's own development strategy of "convergence and coordination of many human wills and actions" (Hirschman 1958, p. 147). It might be thought by many in the aid community that adding 'fake money' won't help communities have more resources. However, what is more important is what a community does, and what it becomes as a result of what it does, which is sometimes called 'learning by doing'.

Both Allyn Young (1928) and Albert Hirschman (1958) in their seminal works on economic development emphasized external returns to scale and the diversification of inputs that are induced with specialization: as firms specialize, economies diversify. In his book *The Strategy of Economic Development* Hirschman (1958) argues that once an industry causes other industries in the supply chain to be established locally, its chances of survival and continuous expansion will be significantly better. Both private and social gains grow exponentially as regional industrial clusters take root with backward and forward linkages. These are defined by Hirschman as follow:

- backward linkages: demand-side stimuli, formed when a new activity, e.g., import replacement, generates effective demand for local inputs, making viable (new) upstream activities that feed into it by supplying inputs.
- forward linkages: supply-side stimuli formed when new activities emerge downstream as a result of a plant or business coming into operation, and intermediate goods become available for local producers that previously did not exist.

"That growth *around* a venture creates favorable conditions for the health and growth of that venture can hardly be doubted" (Hirschman 1958, p. 134) Hirschman cautions that in the less and especially the least developed regions, it may take considerable time before

'lonely outposts' are joined by other firms through *linkage effects* (Hirschman 1958, p. 135). Hirschman was adamant that imports served a purpose, especially as input into infant industries. As well, imports could awaken industrialization pathways: "imports fulfill the very important function of demand formation and demand reconnaissance for the country's entrepreneurs" (Hirschman 1958, p. 123).

Jane Jacobs (2000), a strong proponent of CCs, puts even more emphasis on the potential benefits of certain types of imports, especially those that feed into production, as incoming bundles of embodied knowledge and know-how, offering a stepping stone towards local innovation, stimulating diverse multi-staged products with a significant part for local use through backward integration, to produce previously imported inputs, and "unexpected recombinant matings with nearby processes and products to produce novel 'offspring' and forward linkages" (Ellerman 2005). She described economic development as expanding diversity and "exuberant" import replacement. Critical of the static efficiency of greater specialization, she emphasized the dynamic development from the branching off of new kinds of work, great inter-sectoral interaction and diversity in production firms, goods, and skills.

Hausmann and Hidalgo (2010, 2011) and Hausmann et al. (2013) have used network analytics to draw out empirical stylized facts that support these theories. They show that the richest countries are those with more complex economies, with the greatest diversity of export goods production. Products are made by combining specific subsets of non-tradable productive inputs, which they call capabilities. The more diverse the capabilities in an economy the more likely is the production of specialized products (that is, products that are less ubiquitous), which can sell at a premium given their lower competition. In a feedback loop, countries that have more capabilities are more able to make more specialized products, that is, are more diversified. Hausmann and Hidalgo's model implies that the return to the accumulation of new capabilities increases exponentially (even factorially) with the number of capabilities already available in a country. This conclusion leads the authors to warn of a *quiescence trap*: countries with few capabilities will have negligible or no return to the accumulation of more capabilities. Since these cumulative network effects have such a strong impact that reinforce and multiply existing regional inequalities, it is incredibly important that mitigating solutions are found. CC systems have the potential to localize and leverage product synergies in a manner that, among the authors mentioned in this section, only Jane Jacobs recognized.

The emphasis by both Hirschman and Jacobs on the importance of imports as essential for dynamic growth raises concerns with the pure mutual credit CC in the periphery of towns and communities in less developed countries. These communities generally lack the diversity in 'productive knowledge' and start out dependent on aid and imports (goods and services from outside the community). The typical mutual credit has a strict non-conversion policy, and a policy of debtors equaling creditors. Albert Hirschman's criticisms of the balanced growth arguments for developing countries (Hirschman 1958, pp. 50–74) can be extended to pure mutual credits. Hirschman believed in "development as a chain of disequilibria". His view is counter to the pure mutual credit modality where the introduction of CC through trade is assumed to converge upon a stable equilibrium growth path of reciprocity. By its very nature, uncoordinated and disenfranchised individuals in a peripheral economic area, in Hirschman's words, will find it difficult "to marshal sufficient entrepreneurial and managerial ability to set up at the same time a whole flock of industries that are going to take in each other's output! . . . In other words, if a country were ready to apply the doctrine of balanced growth it would not be underdeveloped in the first place". (Hirschman 1958, pp. 53–54).

This critique of a mutual credit as an aid modality, does not mean that the optimal CC system modality is a freely convertible CC. Making a CC liquid by allowing free conversion defeats the purpose of the CC's existence which is that it 'must be spent' (see Amato and Fantacci 2014, p. 120). The GE case study in the next section highlights the tension and challenges between these two opposing principles of liquidity versus clearing. We

locate the GE CC system as a hybrid aid-CC-cash-voucher ecosystem and an interesting case study for donors' foray into a CC systems. Its evolution, particularly during the COVID-19 period, offers a lesson that a CC must be managed as a system, and not only at the level of individual recipients, unlike the CVA modality. GE transaction data suggests that despite suboptimal membership policies in 2020, diversity in 'productive knowledge' across active members is growing and that CCs appear to offer a robust and powerful tool to localize and leverage donor aid for inclusive economic development in peripheral low-income communities.

## 4. Community Currencies in Kenya

According to a Kenyan Central Bank (2019) household survey it is estimated that in this lowest wealth quintile, 22% were excluded from accessing financial services, and instead kept money in a secret hiding place and borrowed from family and friends in 2019. Add another 15% who had access to informal services like savings groups (chamas), shopkeeper credit, money lenders or employer credit. In 2019 two thirds of all households surveyed would have incurred a problem during their income cycle of not being able to meet their daily expenses, and 30% of the time they would go to their *chama* (informal savings group) for help. In this same study it was estimated that mobile phones were owned by 91% of people 16 and over, 79% used them for making payments. Despite the meteoric rise in using mobile phones for payments, with the introduction of Shilling M-Pesa by Safaricom in 2007, cash is still the primary tool for daily expenses. That said, 71% of all mobile phone users had used them to pay an amount that was less than 2.50 KSh each month.

GE is a foundation focused on supporting the economy of people in the lowest wealth quintile in Kenya. Its founder, Will Ruddick, had decided on using CCs to prove a thesis— that it was not from a lack of resources that stopped people from meeting their daily needs. Indeed, he saw in the slums of Mombasa and Nairobi unemployed manpower and spare capacity everywhere. What was not in supply was money. He found that by introducing a new alternative form of local currency he could promote better coordination among local traders and pathways for economic progress. Below is a history of the methods in which he distributed a community CC in the slums or informal sectors of Kenya and how it has become an ingenious humanitarian- development-aid tool.

### 4.1. A Short History

In 2010 Ruddick and a Mombasa-based charity Koru Kenya, that later became Grass-roots Economics Foundation, implemented a paper CC called *Eco-Pesa* (pesa in Swahili means money) that was distributed as a *convertible CC* into three villages, informal urban settlements, in Kongowea, Kenya of 2000 people[10]. Modeled off the Berkshares, the CC was fully backed by donor funds, and convertible with monthly limits. Seventy-five local businesses received a small allotment as encouragement and received training in accepting and spending CCs. Businesses could buy more at 20% discount and redeem into KSh for a 20% fee, to incentivize circulation and provide collateral. After one month these fees were dropped and it became fully convertible but with quantity and timing limits. A donor focused on waste collection and tree planting, paid people in CCs for such tasks, and created youth-run businesses for the purpose, backed by KSh funds. Businesses mostly used their Eco-Pesas to pay for these assorted environmental services, helping to circulate the currency. By the time the currency wound down through the conversion back into donor KSh after 7 months, 20 tons of waste had been collected, thousands of trees planted, and businesses surveyed had a 20% increase in sales (for more details see Ruddick 2011). The model showed that local resources could be mobilized and become the bulk of value-added in an environmental development aid project.

Following this successful pilot, from 2013 until 2017, Ruddick et al. (2015) and Koru Kenya introduced a new CC (called Bangla-Pesa) in Bangladesh, Mombasa's largest slum. In this case the CC had no national currency backing and no conversion, rather it was run

through the mutual cooperation of local businesses to honor the currency. Initially around 100 rising to 200 businesses (each requiring four other business guarantors) received a one-time payment of 200 Bangla-Pesa (equal to the average budget for a family's daily needs, US$2 at the time). Additional pesa was given to a community fund for their spending (for details see Dissaux and Ruddick 2017). Each member promised to accept the community currency as much as they spent it which conforms with the principle of clearing and reciprocity. The community businesses voluntarily pegged the CC value to KSh 1:1 in the marketplace.

What made this system similar to a mutual credit, despite a positive starting position and minimum balance limit of zero, was its non-convertibility and an expiration date on the paper notes one year after issuance. All members who attended the end of year cooperative meeting, could have their pesa stamped for renewal, but only up to their initial allocation—200 Bangla-Pesa. All remaining Bangla-Pesa in circulation that were unstamped were effectively cancelled, which included all the balances of those who did not attend the meeting. This act crudely enforced the principle of clearing and balance. While these expungements brought the CC system back towards equilibrium (original members all holding 200) it was far from complete. There would remain significant redistribution of welfare even with this act—*from* those who held Bangla-Pesa above 200, *to* those holding less than 200 or even zero (business guarantors were backups for accepting pesa in exchange for goods, but not for payments to bring zero balances back to 200).

As with all mutual credits those who spent more than they earned imposed 'excess' positive balances (amounts above 200) onto other members and a few in particular.[11] In attempts to promote circulation and avoid these unfair wealth transfers, GE "field-officers . . . had to spend a good deal of time simply going from one business to the next, finding out where the currency accumulated, explaining the importance of spending, connecting businesses that hoarded with those that couldn't get Bangla-Pesas, or suggesting new ways to develop possibilities to spend and earn the local money. The task was made more time-consuming because the paper-based nature of the currency made it difficult to trace its circulation paths" (Barinaga et al. 2019). These mutual credit CC systems expanded to encompass six different communities across Kenya all in extremely low-income neighborhoods, some rural but mostly urban areas, around Nairobi and Mombasa: Bangla-Pesa, Miyani-Pesa, Zeni-Pesa, Gatina-Pesa etc. In 2015, a small survey suggested that over 80% of users felt that sales had substantially increased due to the use of CCs (Ruddick et al. 2015). Validating Hirschman's theory that local resources only needed a "binding agent" to call forth and enlist "hidden, scattered, or badly utilized" resources for development purposes (Hirschman 1958, p. 5). In this case, donor funds were limited to GE cost of operations e.g., the printing of the currency etc.

Despite the overall success, there were still problems. As a mutual credit membership grows imbalances are also likely to grow. To avoid excess positive balances many in the community chose not to participate. The dilemma was often put down to lack of trust (see Dissaux and Ruddick 2017) but it could well be that at its root is the heterogeneity in import propensities, or in other words the ability to spend locally, is not the same for every business. Vendors only benefited if they "could source their stock locally [from those who could also source their inputs locally and so on] . . . Otherwise, businesses would face losses if they accepted too much CC . . . The most popular shops would get inundated with too much CC to use and would have to drastically limit acceptance" (Dissaux and Ruddick 2017). The unbalanced way in which open-production networks grow and progress means that by definition exports from (most) networks will be less than imports needed by networks, and decentralized coordination is not enough to resolve this problem.

In 2017, GE unified all the different paper CCs into one fungible paper CC called Sarafu-Credit (Sarafu meaning currency in Kiswahili), retaining the expiration date and thus the promise of reciprocity (see Figure 1). But rather than purely decentralized production, GE had started in 2016 to invest in community businesses that would guarantee acceptance of Sarafu, which we call producer credit in Table 1. The establishment of general

stores, subsidized by KSh donor funds, located in the communities, provided goods that effectively backed the currency. Donor funds also partially financed cooperative businesses (public goods) like a maize mill and coconut oil factory, that would always accept CCs. With these 'authorized entities' GE could manage their excess balances through con-version into KSh, and thereby manage CC supply, system wide and reduce overall imbalances.

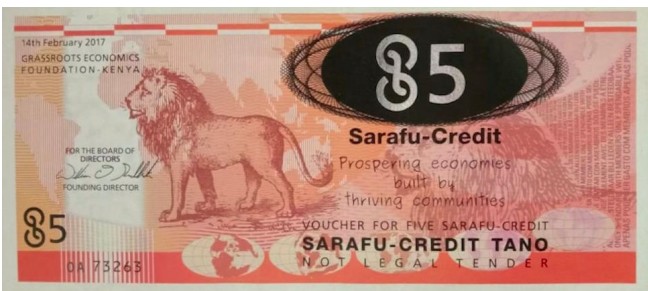 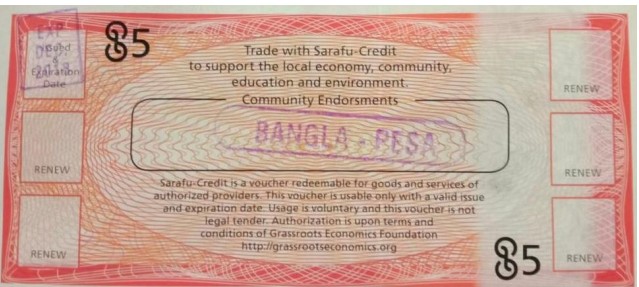

**Figure 1.** Sarafu Paper Currency, 2017. This particular currency note was renewed by a Bangla-Pesa community stamp. Expiration was a way to cancel excess positive balances and incentivize reciprocity.

**Table 1.** Overall Timeline and Brief Evolution of GE CCs and Policies.

| Approx. Dates | Regist. Users | Type | Token(s) | Main Actors & Technology | Official Exchange to National Currency |
|---|---|---|---|---|---|
| 2010–2011 | 100 | Convertible Aid + Payments | Eco-Pesa Single Currency | GE, Paper money | Convertible into KSh (monthly limits) |
| 2013–2017 | 500 | Mutual Credit Promise of reciprocity | Bangla-Pesa + Multiple Local Currencies | Community Business Groups + GE, Paper money with expiration date | Not convertible into KSh. Reciprocity enforced through cancellation of the currency after one year. |
| 2017–2018 | 4000 | Producer Credit Promise of reciprocity | Sarafu-Credit Single Currency | Community Business Cooperatives + GE, Paper money with expiration date | Cooperative Business Profits and Donor Aid used to purchase selective 'excess balances' |
| 2019 | 20,000 | Producer Credit + Aid (airdrop) | Bangla Token + Multiple Digital Tokens | Business Cooperatives + Chamas + GE, Blockchain + Bonding Curve | Select vendors could sell a percentage of their balance with an upper limit. Conversion with a bonding curve safeguard with M-Pesa Agents. |
| 2020–2021 | 30,000 | Producer Credit + Aid (airdrop) + rewards/demurrage | Sarafu Single Token | Business Cooperatives + Chamas + GE, Blockchain | Donor purchases in KSh of 50% Sarafu balance from selected chama's (once a month). |

In late 2018 GE began transferring the paper currencies to digital records accessible on mobile phones and by 2019 all of the Sarafu became digital currencies, operating on the POA blockchain public ledger. Payments could be done without the internet on simple mobile phones and donor funds paid all the telecom fees (Figure 2 shows an example of mobile phone use for the 2020 Sarafu system). GE made use of the Bancor protocol[12] which allowed GE to return to multiple CCs or crypto tokens: Bangla, Miyani, Zeni, Gatina, etc. An ordinary membership would only allow one community token per digital wallet, and only one registered wallet per mobile phone. Exchange rates between tokens were algorithmically determined by a 'bonding curve', with the goal of stabilizing growing or

falling inter-community trade deficits. The Sarafu token was the reserve currency that acted as the settlement token for inter-community trades, and its access was limited to exchange rate mediating entities, just like Keynes' restriction of Bancor to central banks.

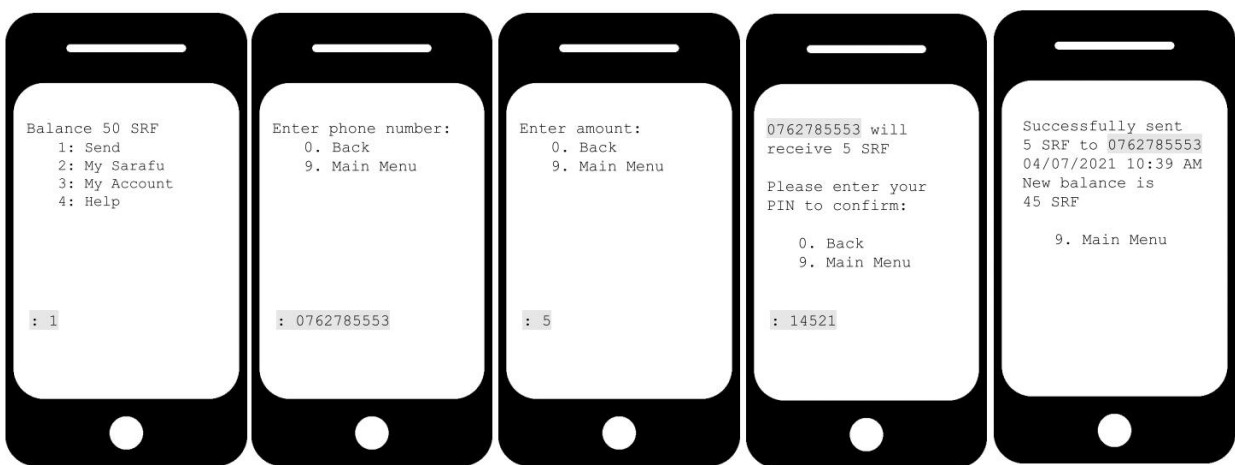

**Figure 2.** The 2020-21 Sarafu Network Mobile Phone Menus. The network allows trading CCs without an Internet connection. This example shows a user sending five Sarafu tokens to another user, after which both parties would receive a receipt SMS for the transaction. A similar menu was used from 2018–2019 although at that time it was a community token (either Bangla, Miyani, Zeni, Gatina etc) not Sarafu that was transferred by standard individual account holders.

The expiration date on CCs was removed and the CC had morphed away from a mutual credit and the promise of reciprocity, towards tokens that represented outright CC transfers with no personal clearing obligation.[13] Gone too were the business guarantors. All new member accounts were now seen as aid recipients first, with an initial allotment of now 400 community tokens (approximately US\$4). Free riders—one off aid recipients that joined, spent, and left—would create a buildup of excess positive balances for those that remained.

Selective conversions remained an important way to manage supply. GE and donor funds financed profitable cooperative businesses that could use their profits to reduce the supply of CCs: water tanks, car washes, phone charging stations, gutter installation. Around 10 designated vendors with a net positive CC inflow due to popularity and insufficient local suppliers were offered conversion opportunities. Donors too could always inject Shillings into the local community helping to reduce excess CC balances and maintain member satisfaction.

At first blush it appears that with discretionary conversions and a bonding curve that imposed its own capital controls through price adjustments, this method was far more elegant in leveraging aid money and managing imbalances than the brute force expiration of paper pesa. In addition, anecdotal evidence suggests that non-financial transactions were not pricing in the bonding curve exchange rate. Instead, most traders in real goods and services treated their community CC token as if it were still 1:1 with Kenyan Shillings, perhaps avoiding the negative financial impositions from exchange rate fluctuations on the real side of the economy.[14]

In January 2020 GE moved from the POA blockchain to another (xDai) to save on-chain transaction costs and was forced to drop the bonding curve and its exchange rate calculations. The multiple CC ecosystem was replaced with a single token: Sarafu.[15] All CC accounts were exchanged 1:1 for the new Sarafu tokens, validating community norms regarding exchange rates. One might say that without a bonding curve to lean against incentives for mass conversion, programmatic attention to disequilibrium was now more important.

In early 2020 GE partnered officially with the Danish, Norwegian, and Kenyan Red Cross Societies. In March 2020 the Red Cross began an active CC pilot in Mukuru, one

of Nairobi's largest urban slums, which quickly turned into an improvised COVID-19 response system (Ruddick 2020b). A timeline of regional active membership (at least one trade per month between standard members and chamas) is shown in Figure 3. With a COVID-19 lockdown and impending national economic collapse the Red Cross backed a national membership drive where anyone in Kenya could dial a code on their phones and get 100 credits—equal to US$1 of goods and services among voluntarily accepting members. GE also ran pilots in Nairobi, Mombasa, Kwale and Kilifi, injecting 100,000 US dollars worth of Sarafu with new membership offerings (400 Sarafu each). Later, in January 2021 the Red Cross began another pilot in Kisauni Mombasa.

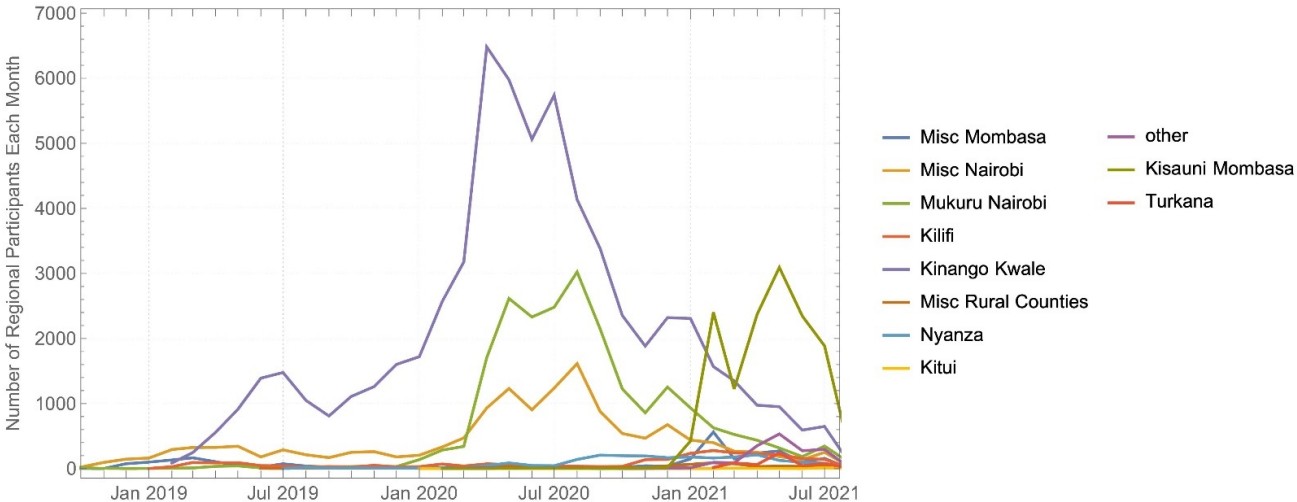

**Figure 3.** Monthly Number of Active Participants By Region. Counted are only members with at least one transaction in the month. The data includes only transactions between CC members and chamas.[16]

These initiatives were in response to the dramatic rise in poverty and unemployment from the COVID-19 pandemic, and the new opportunity for expansion that the Red Cross partnership offered GE. Active membership, the number of users transacting within a monthly period, increased fivefold from 2000 at the end of 2019 to a peak of 10,000 in mid-2020 (see Figure 3 for user activity disaggregated by region). Overall, these interventions were said to be cost effective, leveraging donor funds 10 or more times.[17] In the process of these pilots community members got to increase their sales and make new local trading connections, diversifying the range of local inputs into their production methods. While membership increased by a factor of five, the number of transactions increased by six-fold (see Figure 4).

During 2020 Kenyan GDP growth fell by 0.3% (compare this to an increase of 5.4% in 2019).[18] Since May 2021 the Kenyan government has been lifting COVID-19 restrictions and curfews. The GE CC data in Figure 3 shows that the number of active Sarafu users fell back to 2000 per month by July 2021. Further research is needed to assess the long run impact, but the rise and fall of active members offers possible lessons. It may indicate that during the worst period of COVID-19 and economic shutdown the CC was a substitute for scarce Shillings and starting in October this was reversed because Shillings became less scarce. This might infer that the CC stimulus did its job and was no longer needed as indicated by the steady exit of people using the CC since October 2021. Alternatively, it could be that real side economic development and the creation of trading pathways does not happen through random recruitment of people with a mobile phone, which are otherwise unconnected. This latter hypothesis infers that trading networks grow organically not randomly. If membership is growing faster than the backward and forward linkages needed to embed their presence, then orphaned members will try to liquidate and exit the system, offloading their balances onto others.

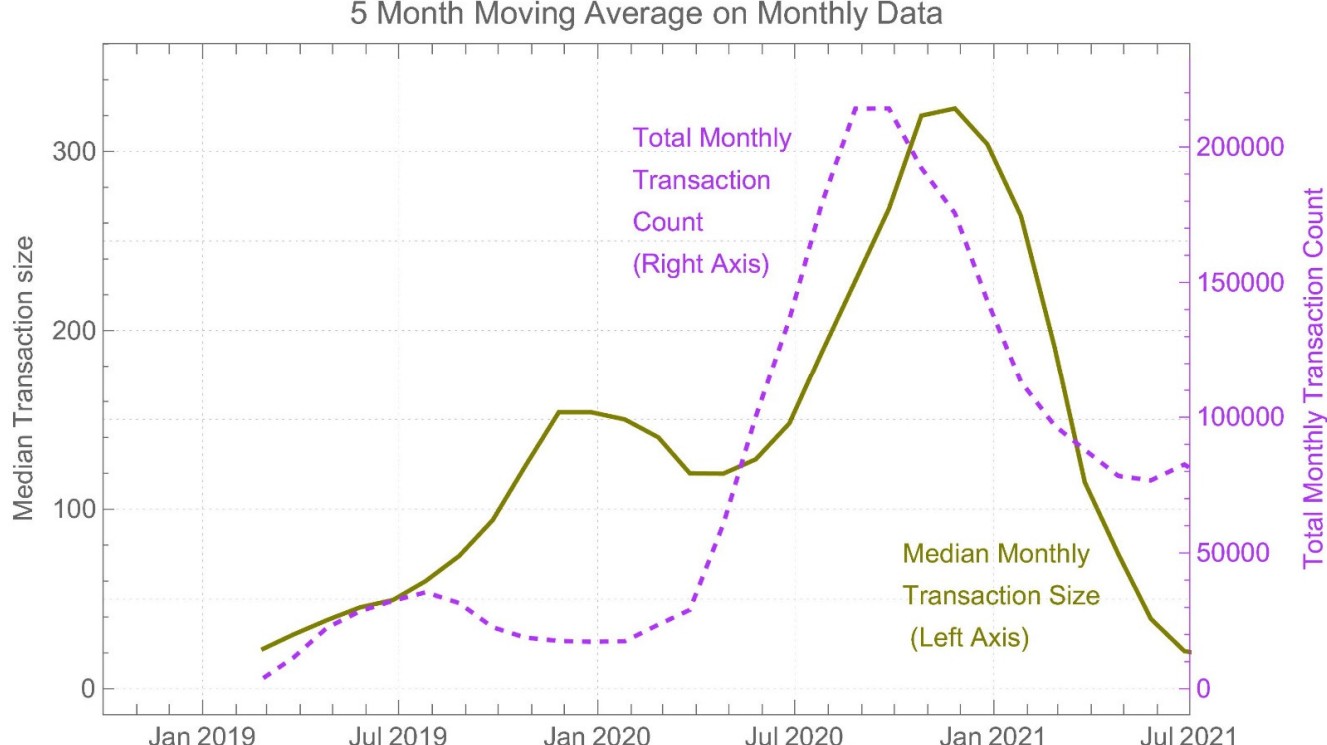

**Figure 4.** Monthly Transaction Count and Median Transaction Size.

End of quarter balances are shown in Figure 5 as 'violin distributions', which are a useful tool for visualizing the tails of a distribution. It would appear that the sudden rise in transient membership and their exit put considerable pressure on excess positive balances. By the end of the first three months of 2020 50% of members had balances of between 100 and 400 Sarafu (between US$1 and US$4), but 25% had more than 400 and some of these accounts were quite large, over 25,000 KSh (approx. US$250). One can imagine a business in a slum area that might regularly accept Sarafu, suddenly having more customers spending with them, and they now have a buildup of Sarafu—they could not spend as fast as they earned. Large balances are an indication that backward linkages have not yet formed. At best the higher balances will constrain their cash flow temporarily while they find new vendors to buy local inputs from, or at worst leads them to stop accepting Sarafu and potentially dropping out of the CC system.

One avenue for excess holdings for most individuals is their savings-group or chama.[19] Figure 5 shows how excess balances moved from individual accounts to chama accounts by the end of 2020. In the first half of 2020 GE tried a pilot of monthly cash purchases of up to 50 percent of chama balances (at a maximum of 30,000 Sarafu or US$300) from community groups aimed to pull 'excess' Sarafu out of circulation and balance the system. Only chamas with high-trade activity were offered this opportunity and can be thought of as a reward for good behavior. Without further research it is hard to discern whether individual vendors transferring their balances to their chama was from involuntary savings (due to the injection of CCs during the COVID-19 rapid response stimulus) or voluntary savings (reduction in spending Sarafu in preference to spend or save Shillings). The latter would be a move away from the primary goal—that a CC is for spending.

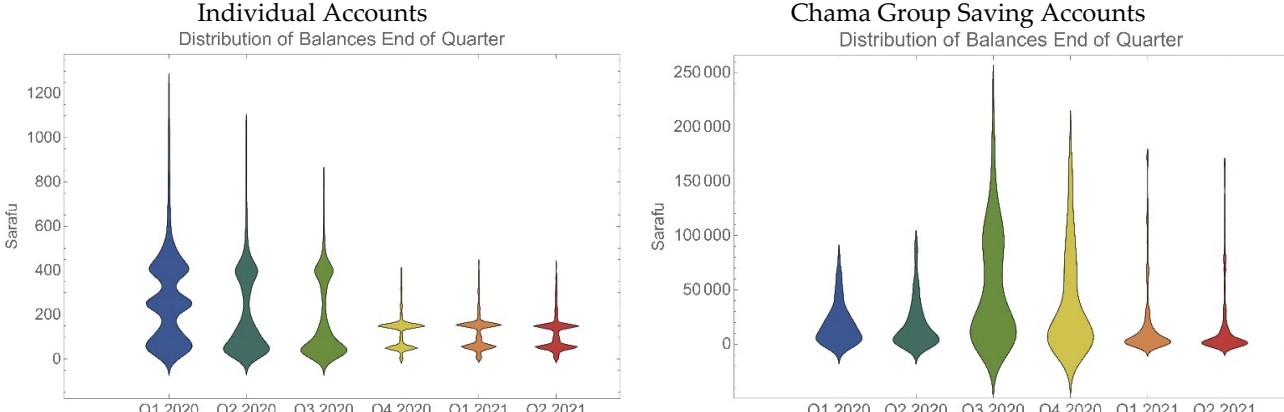

**Figure 5.** Violin Distribution Charts for End of Quarter Balances. Different colors represent different quarters from Q1 2020 to Q2 2021. Data from the xDai Blockchain. Units in Sarafu where approximately 100 Sarafu equals US$1.

In September 2020, to improve upon the chama conversion policy, GE moved away from direct cash solutions for chamas and instead encouraged chamas to select a vendor of choice in their respective communities where they could purchase food worth a maximum of 10,000 Sarafu, twice per month, and then GE would cash out that vendor. This indirect 'voucher' route, seems to be a better way to reduce imbalances as it avoids incentivizing treatment of the CC as a liquid financial asset to be saved rather than spent. By 2021 both individuals and chamas have reduced their extreme tails.

Apart from incentivizing chama trading activity, in October 2020 GE also introduced a monthly fee of 2% on individual balances in an attempt to encourage users to spend faster. In aggregate, the same amount was to be sent back out to users with a higher monthly trading activity. Figure 4 does show that by July 2021, despite active membership being the same as what it was at end 2019 the number of transactions was at least three times higher (red line), and the size of the median transaction had fallen to a new low (blue line). Both trends could likely be in response to these new penalties and rewards. Further research is needed to assess the degree of 'transaction washing' that maybe taking place.

GE also has proposals that involve micro payments in Shillings to reward circulation of Sarafu. Indeed, as a consequence of the Red Cross partnership Kenyan Shillings have become an increasingly important ingredient in helping to manage the supply of CCs, offering a mix of Sarafu or Shillings for development projects, in line with the Red Cross familiarity with Kenyan Shilling CVAs over CCs. Training and capacity development, platform infrastructure, imported capital, community capacity development and community assets like buildings or farms that offer necessary goods have become anchor institutions in the CC network and are often financed in Shillings (green arrows in Figure 6). CC flows along an orange line between the community members, chamas, and social enterprises. This flow is presented as being independent of Shillings since conversions are discretionary. Conversion is with designated parties and is for the purpose of managing imbalances and supply; it is not used to stabilize the value of the CC or its liquidity.

The blockchain allows for data collection that allows impact S hilling investment donors to give cash directly to recipients and social enterprises tied to sustainable development goals. Blue lines in Figure 6 show data shared by users. Community members can be endorsed on the basis of their data, or they can curate their own blockchain data profile to attract impact investors. CC transfers can also be distributed to users as aid or income on the basis of surveys and metrics e.g., on transaction data that proves residence in vulnerable household areas, helping to target future aid and cash transfer programs.[20]

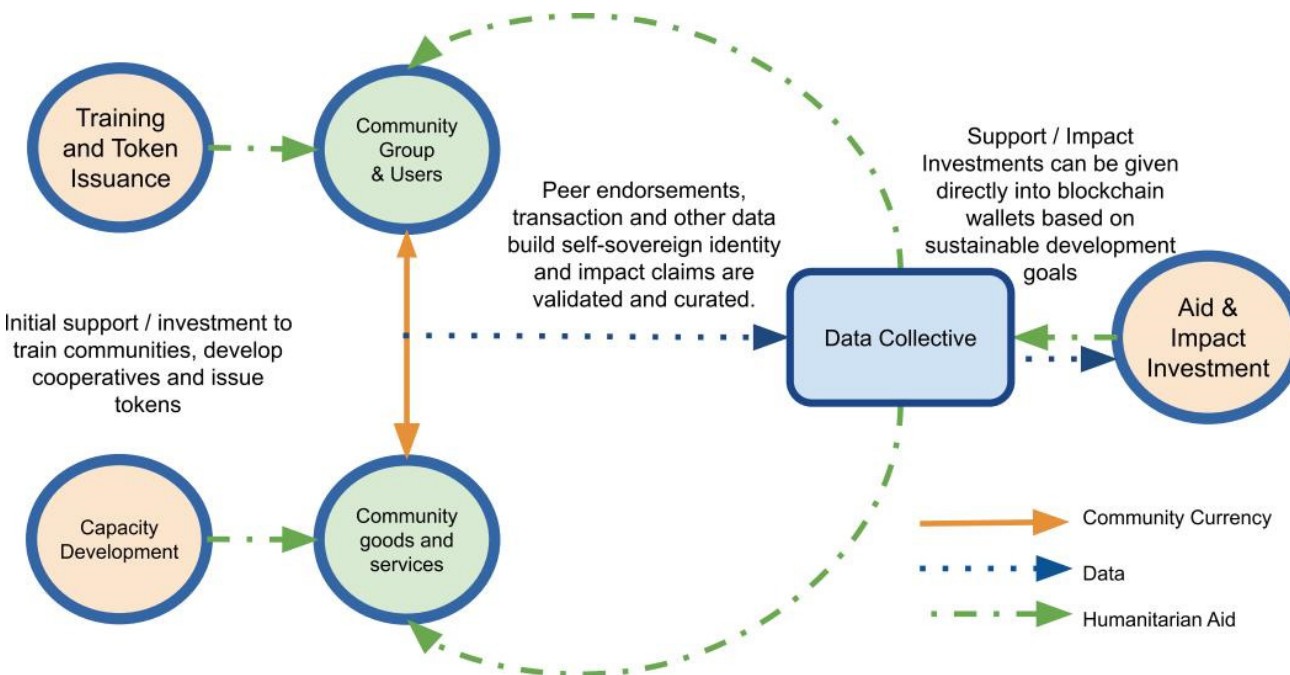

**Figure 6.** Grassroots Economics Complementary Currency (CC) Model on the xDai blockchain 2020–2021.

### 4.2. Real Trade Network

Registered members on the GE platform since September 2018 shows that labor, food, farming, and shopkeepers (shop) are the largest sectors, as shown by the pie chart in Figure 7. It is important to note that there are no data on the exact goods that are transacted in Sarafu. Rather the only information we have is on the seller's member profile, which states what single type of good or service they sell *in general*. There were 236 chama accounts in July 2021, listed here as "savings". It is important to note that chamas, or group accounts, also have business income and spending thus transactions are not just member 'savings'.

A further breakdown within each business sector is a description of what they sell or how they earn CCs, which they describe themselves. We can see from these more detailed breakdowns in Figure 8 that retail shops (duka in Swahili) are vendors that likely own or rent a store, whereas food sellers are producers, intermediaries for delivery, or retailers hawking or putting up a temporary stand on a roadside. Vegetables (or mboga) are the most common food because of the existence of rural communities especially in Kinango Kwale. We can see from the education sector that the Red Cross must be paying (at least partly) in Sarafu their Red Cross volunteers, or training them in their use of Sarafu.

A randomly selected supply chain spanning the first four days of July 2021, versus the prior 30 days leading up to and including these four days, is presented in Figures 9 and 10. Repeat bilateral trades between the same parties are combined into one link, each node or circle reflects a trader. We see that the supply chains are relatively linear over a very short time period (4 days), but quickly turn into circular and complex mappings over longer time horizons.

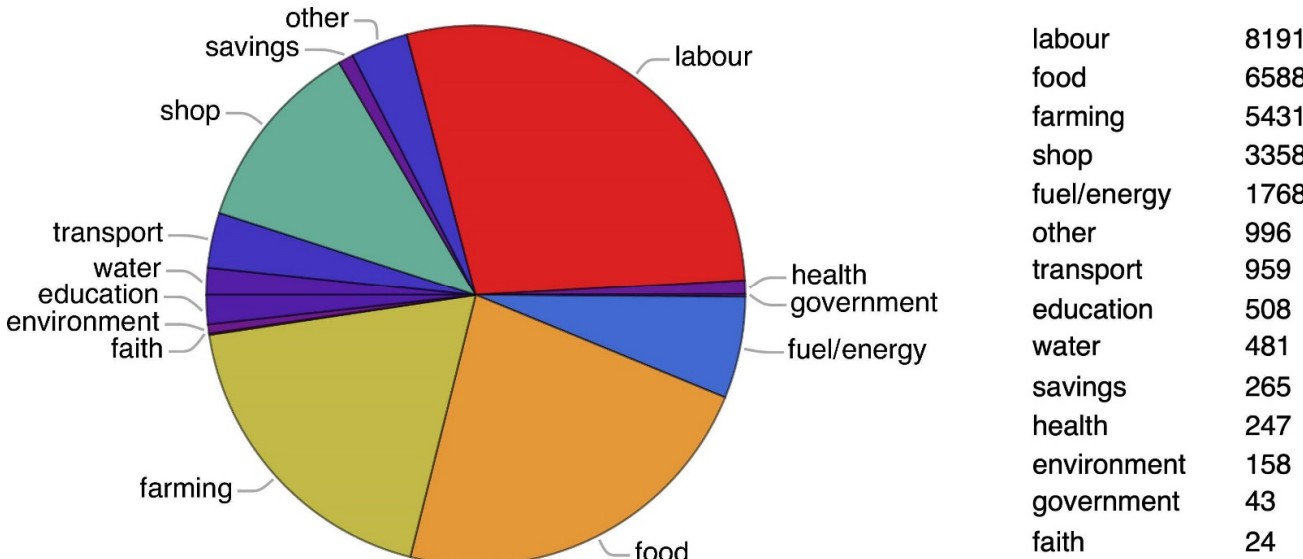

**Figure 7.** Business-type of all registered standard members and chamas since September 2018 to July 2021. Members choose one of 14 Business Sectors when registering. Members can update their member profile if they choose.

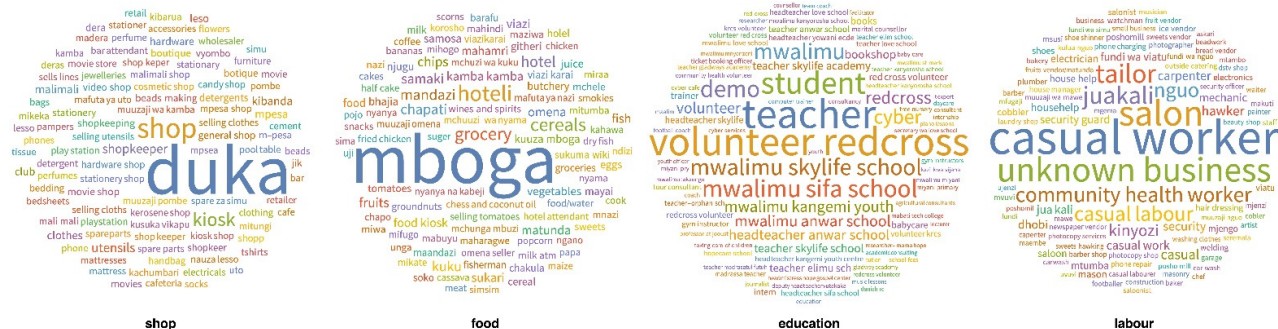

**Figure 8.** Breakdown of Four Business Types into Vendor or Wage Earner Descriptions Chosen by Members. Word cloud is weighted by membership numbers in each sector.

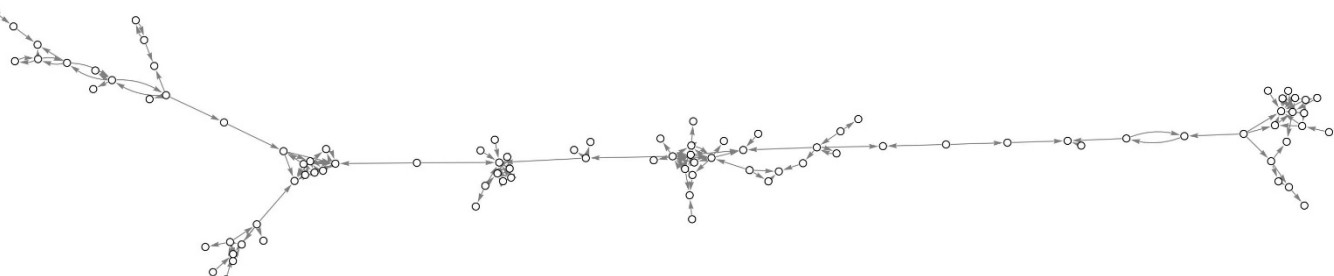

**Figure 9.** Supply Chain for first 4 days of July 2021. Largest component only. Standard member and chama nodes only.

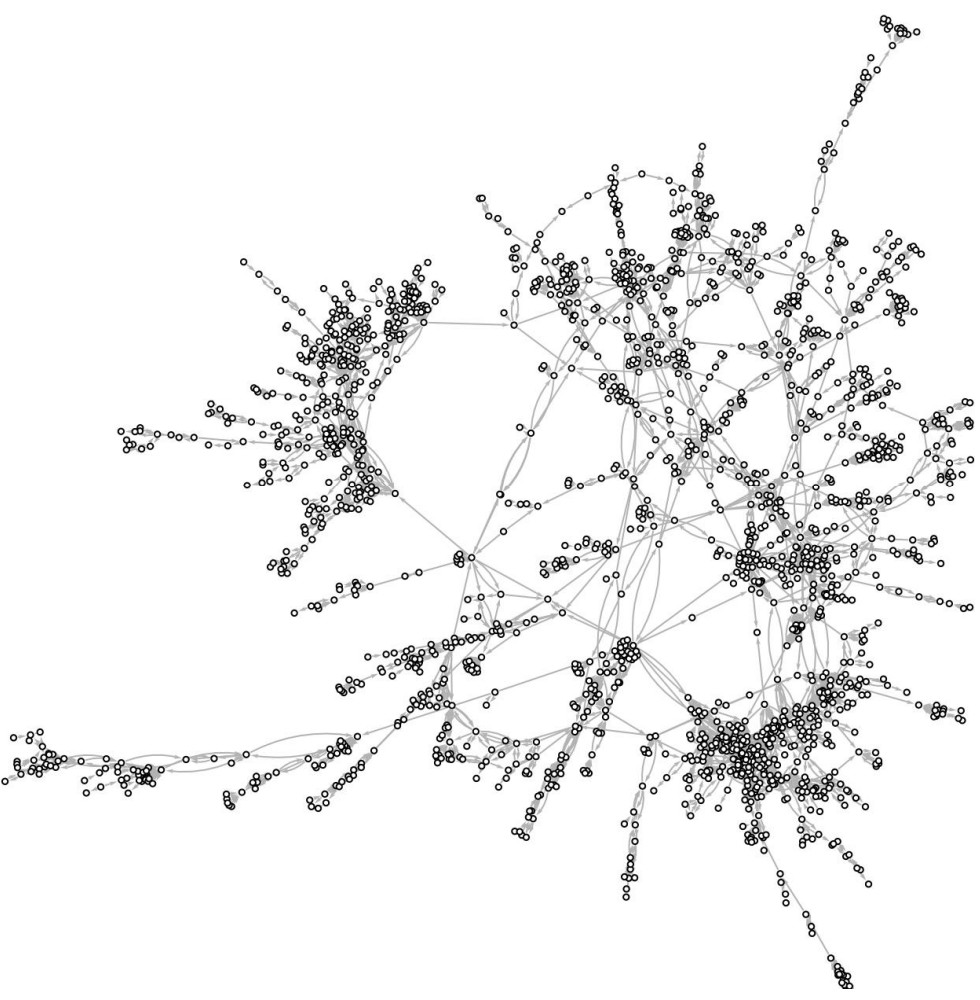

**Figure 10.** Supply Chain for 34 days from 1st June 2021. Largest component only. Standard member and chama nodes only.

A more comprehensive presentation of the network is to aggregate transactions across sectors as in Figure 11. This chord diagram weights total sales and purchases of business-types as an undirected graph according to all member profiles registered since September 2018, excluding disbursements and withdrawals or what we call financial transactions. Internal savings with chamas are included. This offers a visual of backward/forward linkages or both inputs and outputs. Total transactions sum up to US$3.1 million from late September 2018 to end-July 2021. Food sellers, mostly in the informal sector, make up the bulk of users and consist of fruit and vegetable kiosks, restaurants, wholesalers, and in some cases they are farmers selling their own produce. They have the greatest purchasing power, paying wages and buying items from other shops. About one quarter of food purchases and sales are from other food vendors and shows up as the region with no outward linkage. Food vendors also have the most Sarafu going into chama saving accounts. The following quote is from a food seller who is exchanging Shillings for Sarafu from her chama, effectively passing her Shillings onto another member in the group who needs them more, while she uses the Sarafu. The food seller will get her Shillings back at some later date.

> "We always save money in our table banking group [chama] and then exchange it for Sarafu. Everyone gets a share according to their savings. All our phones always have Sarafu which we use for various uses; to buy food and supplies for my pastry business".

—Female, Kinango Kwale

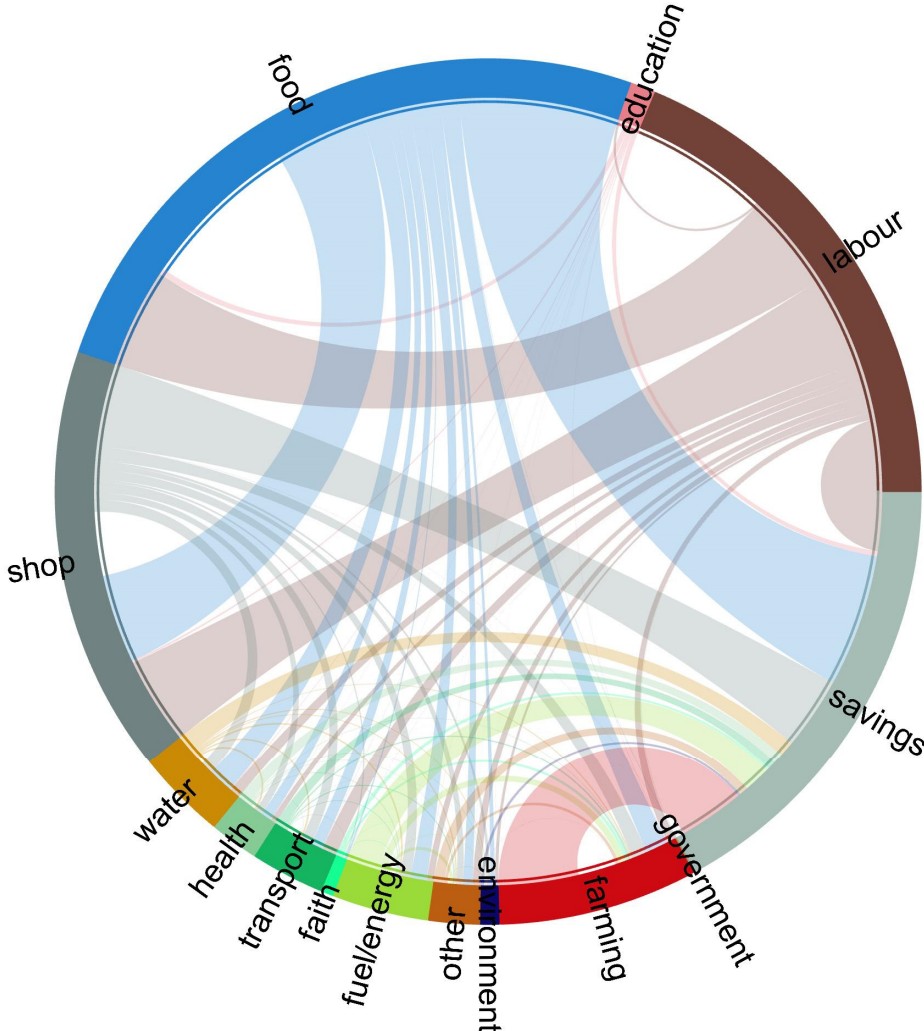

**Figure 11.** Purchases of Seller and Buyer Business-Types, Weighted by Trade Values. All Financial Transactions are Removed Except "Savings" with Chamas. Two-way transfers such as between food vendors purchasing labor, and labor purchasing food, are consolidated as one link.

Following Hausmann and Hidalgo (2011), we map over time the diversity-ubiquity production space. Figure 12 is a series of six consecutive five-month data windows starting the end of December 2018 and ending in June 2021. The heat map represents the number of transactions, normalized for each window. The horizontal axis of each graph is the range of sectoral diversity and sectors are roughly ranked by the number of transactions in each region. The vertical axis is the measure of ubiquity of goods across regions. The number of regions expands over time. Regions are ranked by transactions in the business-type of the first column. A region's diversity is the number of different types of business sectors it contains. A business sector's ubiquity is the number of regions from which goods are demanded/supplied outputs/inputs of that region (backward/forward linkages). To the degree that regions do not trade with each other, and the production and consumption by a business for a given product is entirely within one region, then the product-space matrix for backward and forward linkages (for outputs and inputs) is identical. Since this is largely true, we only include the backward linkages in Figure 12.

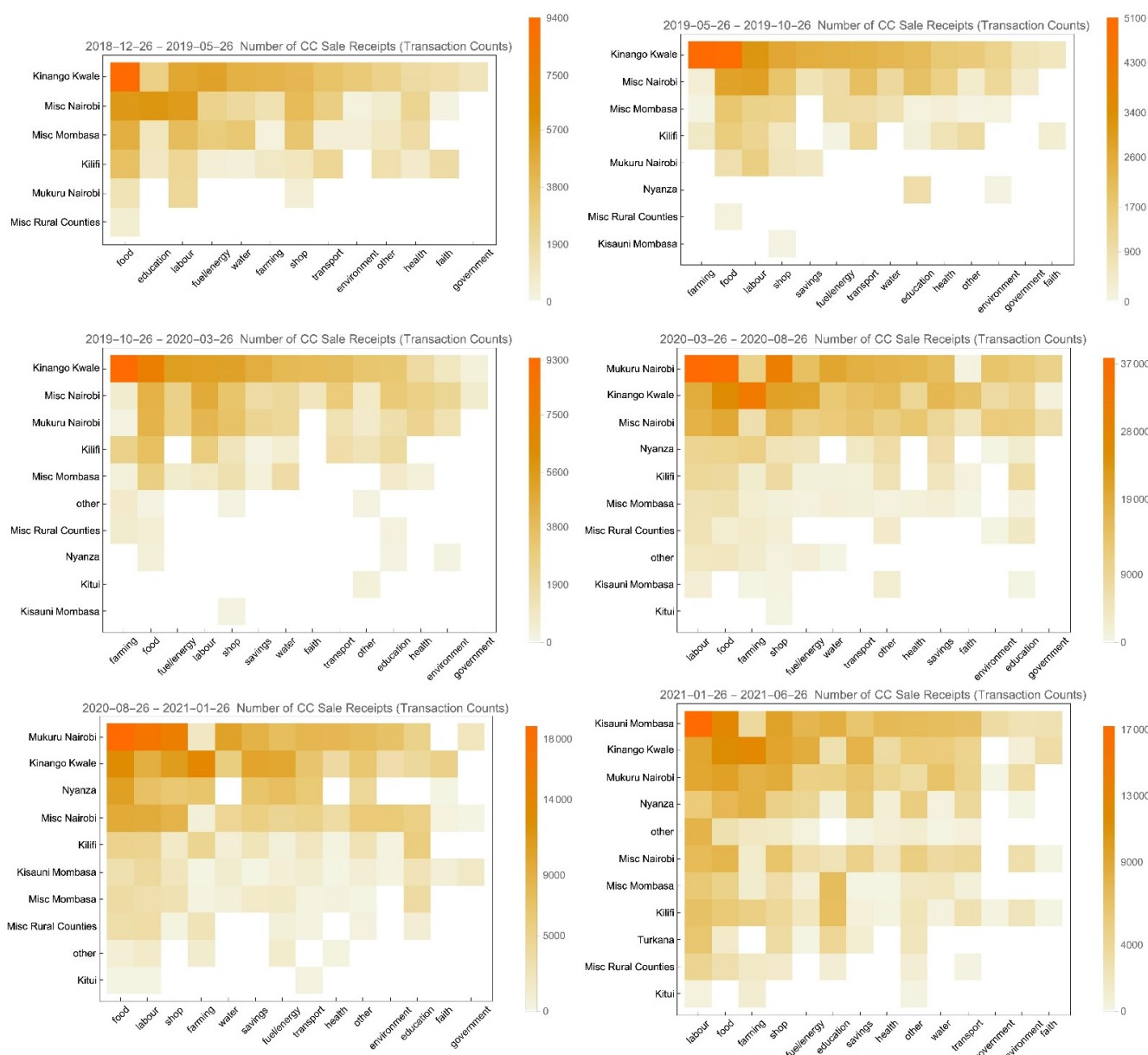

**Figure 12.** Backward Linkages: diversity-ubiquity production space for the entire CC membership. Starting in the first 5 months from 26 December 2018, to the last 5 months ending in 26 June 2021. First row is ranked by sales count per vendor-type by region. Then first column is ranked by sales count per region by the highest vendor-type.

In their study of trade data, Hausmann and Hidalgo (2011) claim to have found a stylized fact of country or regional economic development—a negative relationship between ubiquity and diversity in exports. They also claim that regions that export less ubiquitous goods are generally the most developed regions with the greatest volume of trade and the higher diversity of goods across sectors. These regions are at the top of our vertical axis. According to Hausmann and Hidalgo, regions that export only ubiquitous goods and have less diversity of production in their economy will also be those regions with the least economic progress.

If our data were in accordance with Hausmann and Hidalgo's stylized fact, and if the number of transactions were a sign of economic development, then we would see a triangle in the top left corner of our matrix plots in Figure 12, that goes from high to low intensity as we move down and right. Visually it appears that there is this relationship

although more robust methods would be needed, comparing other measures of regional economic progress, independent of these formulations.

What is apparent is that over time regions with the most transactions do have the greater sectoral diversity, and the volume of trade is across all sectors. The most ubiquitous sellers or products are food, labor, and farming (as previously stated, due to data limitations we assume that sellers only sell one product). According to Hausmann and Hidalgo's methodology, Kinango Kwale and Mukuru Nairobi would be seen as the more fertile development areas because of their diversity of sectors, also in line with Hirschman's view of economic development. This is the opposite of a thesis of comparative advantage where specialization in a single sector cash crop is thought to be a sign of economic progress.

Mukuru Nairobi and the miscellaneous grouping of Nairobi locations are urban locations that also have diverse and deepening backward linkages over time. In contrast the miscellaneous rural communities that are unconsolidated by region do not have dense local networks of trade and depend on purchasing power of labor and food sectors, with little else. Despite this bloc being measured for some time, this grouping has not increased its diversity of sectoral capacity, nor significantly its volume of trade across sectors.

What is most interesting about this sequence of windows is that in the last 5 months all regions continue to become more diverse despite lower volumes of trade and declining membership over the past 5 months, shown in Figure 4. While further research is necessary, this upward trend of greater diversity seems to bode well for economic autonomy and greater resiliency among those that remain in the CC system, post COVID-19.

According to our application of Hirschman's reasoning, induced local demand through the institution of a local currency likely sets in motion not only the increase in total sales of final goods by vendors in low-income communities, but the creation of new and enhanced linkages, an increasing set of capabilities that can take on new initiatives, and learning-by-doing. If our assumption is true that Shillings are quick to leave the periphery through imports then we would interpret this growing diversity as a reflection of replacement and shifting of imports à la Jacobs (2000).

The importance of localizing, decentralizing and democratizing aid in support of collective needs builds community cohesion and stimulates local inclusive economic progress. The building of local 'productive knowledge' and capabilities across diverse product networks with CCs leverages donor aid, for a bigger long run impact for impoverished communities.

## 5. Conclusions

With the adoption of the Agenda 2030 Sustainable Development Goals (SDGs) in 2015, and the 2016 World Health Summit, there has been a growing recognition for a broader humanitarian mandate and better connectivity between humanitarian and development aid. This has brought into focus the need for aid to not just meet individual needs, but to reduce risk and vulnerability over the longer term. One answer is the move from supply-side aid that is in-kind or service based, to demand side oriented aid like vouchers and, in particular, cash. CVA is said to be a market-based approach that is more democratic and can decentralize and flatten humanitarian pursuits that support local markets as opposed to in-kind aid which brings resources in from outside and is top-down. This paper argues for a fourth aid modality, CCs, which are also demand side.

Diverse and densely nested local supply chains have the strongest correlation with economic growth—more relevant than competitiveness, governance, education, and financial depth (Hausmann and Hidalgo 2010). What is more intrinsic to local development is what a community does, and what it becomes as a result of what it does, which is sometimes called 'learning by doing'. When an enterprise or entrepreneur combines different capabilities or inputs they diversify their outputs and meet new needs of other entrepreneurs. Following Hirschman's development strategy, the authors propose that rather than scarce resources as the intrinsic constraint on economic progress, it is a deficiency in utilization and coordination capabilities. CCs can act as a "binding agent" to call forth and enlist

"hidden, scattered, or badly utilized" resources for development purposes (Hirschman 1958, p. 5). CCs are superior in localizing, decentralizing and democratizing aid in support of collective needs. They can be used alongside CVAs to mobilize different actors working towards collective action, building a network that is more resilient and less vulnerable to crises.

Regions and communities that qualify for humanitarian-development aid are in the periphery and can be characterized as leaky buckets. The net outflow of human capital (brain-drain) and official money (via imports and capital flight) generate commodity-, debt- and poverty-traps. These structural factors create an ecosystem that has few capabilities, limited differentiation in products, short and orphaned local supply chains, and thus big holes in the bucket. CVAs, tied to national money, that land into leaky buckets dissipate quickly, due to the high import propensities from the inferiority and gaps in local supply chains. This is part of the *quiescence trap* discussed by Hausmann and Hidalgo (2010, 2011) that limits local spending on local goods and services, even with CVAs. CCs are ways to mitigate this problem by localizing and leveraging product synergies and building embodied knowledge and know-how. Hirschman (1958) calls this development the creation of backward and forward linkages.

The paper outlined a spectrum of CC design that spans from mutual credits without convertibility to freely convertible CCs which can always be sold into the national currency at a fixed price. While still controversial, most CC experts see mutual credits with strict non-convertibility as superior for local economic development and resiliency programs, based on the principle of reciprocity and import substitution (Amato and Fantacci 2014, pp. 118–44; Greco 2018). However, these same experts implicitly assume that the mutual credit equilibrium, where supply equals demand, exists and is stable. That is, individual accounts have a tendency to return back to their starting position in the 'long run'. They also imply that this equilibrium is not balance-of-payments-constrained, or in other words, the imposition of strict import substitution or non-conversion, does not constrain demand and restrict the CC economy from growing. Indeed the opposite—these experts argue that only by limiting imports to the official currency will a CC system grow in terms of backward and forward linkages. Both these conditions of a stable equilibrium and zero imports are compatible and necessary for a CC system with a balanced growth path under a mutual credit with zero convertibility.

Inspired by development theorists like Albert Hirschman and urban economist Jane Jacobs, this paper offers a counter proposition of unbalanced growth. The task of a CC is to keep money local and support local innovation towards "exuberant import replacement" (Jacobs cited in Ellerman 2005). Import substitution is a process, not an end point. If a CC system includes a community of heterogeneous producers, where membership is voluntary and open, then a well-managed system must take account of inevitable imbalances, due to different propensities to import, mismatches in the production system and involuntary free-riding. Imbalances will not naturally converge back to their initial starting point for all participants and it is not easy to simply add participants to fill gaps or ease bottle necks. Complex systems and network analysis tells us that all nodes are interdependent. Hence removing, or adding, nodes from, or to, a network as a treatment for imbalances will have unpredictable outcomes. Vendors without perfect foresight and common knowledge will get stuck and cannot simply jump to the equilibrium—if one even exists. Imports and their substitution should be seen as stepping stones along a path of dynamic expanding "unbalanced growth" (Hirschman 1958). Like medicine, which can be addictive or poisonous, imports should be recognized as a (transient) joint input to production and accommodating them is appropriate for a growing CC system (rather than leaving them to be accommodated by residual national money which is always lacking). Since debt is inappropriate for a poor demographic, CCs as aid, vouchers as the conversion for excess CCs, and donor cash-aid as the conversion for vouchers is a possible hierarchy of aid modality that can remove excess positive balances and leverage cash-aid. It is critical to note that conversion of a CC does not infer freely available conversion. Conversion controls

must be imposed to limit CC liquidity, such as conversion only with authorized entities, only into vouchers, with quantity limits, with time limits etc. Incentives in using the CC must be based upon a principle of reciprocity between CC members, not be replaced by a principle of liquidity—where the CC is treated as a commodity to be converted rather than to be spent. To walk this tightrope of maintaining reciprocity over liquidity with discretionary conversions requires good management, data analysis, and agility in tailoring CC design to fit community needs. Along with their bespoke nature and need for greater management, CCs can empower communities that want to regain control of their local economies and build agency, not only receive individual aid but receive aid that can be leveraged for the entire community.

To examine the use of CCs in the context of humanitarian and development aid the paper took a case study from Kenya where one-in-five citizens live in urban slums. Grassroots Economics (GE) is a foundation that partners with various humanitarian aid donors to distribute CCs in urban slums and rural areas mostly around Nairobi and Mombasa. Since their first pilot in 2010, GE CCs have had a hybrid design located in between a pure mutual credit and a freely convertible CC. Two features of GE CCs that are very different from most other mutual credits, is that no member account goes into debt (the lowest balance is zero), and that supply of CCs is exogenously issued through new member registration. Both modifications can still be thought of as compatible with a mutual credit with open membership where a large number of members spend but do not earn back. What is actually different in the GE design, as compared to most mutual credits, is the willingness to treat these free-rider transfers as aid (or debt forgiveness) and promote reciprocity by other means, including conversions to those that suffer from involuntary CC savings. Despite the differences between the GE hybrid CC and more standard mutual credits, it is our hypothesis that discretionary conversion will be necessary in a community faced with unbalanced growth. Used sparingly or in limited way, conversion reduces excess imbalances that otherwise place a burden on the 'best' members in the system (those that accept CCs in exchange for goods and services) and risk their withdrawal from the CC system. In sum, we agree with one of our anonymous referees that "the strict mutual credit is not necessarily always the best solution, although I would argue that it is always likely to be a part of the best solution".

The GE framework offers a sandbox to study the use of CCs in a growing and unbalanced mutual credit where aid acts as the equilibrator. The aid buffer offers safeguards for those that volunteer and participate in the CC system, even when experimentation in CC design doesn't work as expected. Our rudimentary case study of GE trade networks show that even with a failed membership drive during COVID-19, where most of the 8000 active new recruits proved to be temporary, the trading network among established members was strengthened. All regions within their CC trading circuits, became more diverse and developed more capabilities. This despite an 18-month travel lockdown, negative GDP growth in the national currency economy, growing inequality, and shrinking opportunities in slums and informal sectors in terms of national money. Further research is needed to properly assess this bright spot using a CC aid modality, during this period of mass disruption and marginalization.

Perhaps a crucial contribution of the GE experiment is that it has utilized an open source blockchain platform since it went digital in September 2018. Due diligence in terms of anonymity and data privacy protects users (Ruddick 2021) while transparency of network connections on a public blockchain provides immutable and verifiable data that researchers can use with confidence. Open and collaborative research to study monetary management of a large-scale CC ecosystem is invaluable. A CC system is intrinsically an accounting system. Data means that researchers can compare notes on different world views using the same data source, and donors can monitor, assess and target future aid in the form of CCs, national money, or vouchers. Data, accounting dashboards, and the cross fertilization between CCs, CVAs, and in-kind aid, should lead to an ever more flexible

toolkit for promoting backward and forward linkages, monitoring economic progress in the real economy, and supporting SDGs.

In sum this paper argues that the humanitarian aid community should study the opportunities that the CC aid modality offers, in combination with their sister CVAs. CCs, like vouchers, can be designed to address local needs. The use of a CC system is less about giving aid to individuals, and more about empowering communities, mobilizing their existing resources, and building new productive capacities through collective action. Unbalanced growth and net imports will characterize economic progress of backward and forward linkages in regions that qualify for humanitarian aid. Donor funds of cash-aid is necessary to support the management of imbalances in the CC system through discretionary and controlled conversion. With the rise in digital currencies, the use of blockchain networks, automated smart contracts and big data will be critical in successfully managing these complex systems. Strong protocols and supporting organizations, like Grassroots Economics Foundation, that are in tune with the shifts and turns of the monetary ecosystem are needed. In this way donor funds can support local economic development and offer a pathway to independence from aid.

**Author Contributions:** Conceptualization, investigation and writing—original draft preparation, review and editing—by L.U., L.E., G.M.G. and W.O.R.; data curation and methodology by L.U. and W.O.R.; data visualization in Mathematica software v.12.3 by L.U.; manuscript funding acquisition by G.M.G. All authors have read and agreed to the published version of the manuscript.

**Funding:** The publication of this paper was financially supported by Monneta (https://monneta.org/en), a German not-for-profit organization, and the Development Economics Research Group at the International Institute of Social Sciences of Erasmus University Rotterdam (https://www.iss.nl/en). Research was funded by the Danish Ministry of Foreign Affairs, grant no. GFIIEG 18-11-CBS.

**Data Availability Statement:** Publicly available datasets were analyzed in this study. The data from September 2018 to July 2021 can be found here: https://www.grassrootseconomics.org/research (accessed on 15 September 2021). The January 2020 to June 2021 subset of this data is also available at a publicly accessible repository Colchester, Essex: UK Data Service. https://reshare.ukdataservice.ac.uk/855142/ (accessed on 5 July 2021). The Data includes pseudonymized account information for around 55,000 users and records of all Sarafu transactions.

**Acknowledgments:** We would like to thank an anonymous referee who provided critical comments and constructive suggestions which greatly improved the quality of the final published paper. All remaining errors are our own. Gratitude is extended to Arash Mahdian at Wolfram Research who assisted in trouble shooting the production of the figures using Wolfram Mathematica software v.12.3 and Claudia Stone for editing the paper. Appreciation also goes to the funders of this research.

**Conflicts of Interest:** Leanne Ussher has financial support from the Danish Ministry of Foreign Affairs. Leanne Ussher, Georgina Gómez and Laura Ebert have no financial relationship with Grassroots Economics. William O. Ruddick is the Founder of Grassroots Economics.

## Notes

[1]   https://www.grassrootseconomics.org/about (accessed on 27 July 2021).

[2]   https://innovation.wfp.org/project/building-blocks (accessed on 27 July 2021).

[3]   https://www.unicef.org/press-releases/unicef-launches-cryptocurrency-fund (accessed on 27 July 2021).

[4]   https://www.unicef.org/innovation/stories/grantee-macro-eyes (accessed on 27 July 2021).

[5]   http://www.stop-hunger.org/home/communautes-locales/laide-locale-au-plus-demunis/lautosuffisance-alimentaire/les-coupons-alimentaires.html (accessed on 27 July 2021).

[6]   In the Berkshires of Massachusetts local banks will sell 100 Berkshares for 95 USD, and buy 95 USD for 100 Berkshares. https://centerforneweconomics.org/apply/local-currencies-program/#BerkShares (accessed on 27 July 2021).

[7]   https://www.community-exchange.org/ (accessed on 27 July 2021).

[8]   https://www.sardexpay.net/ (accessed on 27 July 2021).

[9]   For example the Hudson Valley CC in the United States converts its currency into business advertising in a local newspaper (see Ussher et al. 2019).

[10] The complicated history of GE and its CCs is summarized in this section with a focus on the CC design and its integration with humanitarian donor foundations. More details can be found in the papers cited and on the GE website https://www.grassrootseconomics.org/research (accessed on 27 July 2021).

[11] Regardless of how good intentions might be, such wealth transfers are common among mutual credits that fail to match trades and exclude the option of conversion as discussed above in Section 2.3. This is true regardless of whether they begin with a balance of zero or some positive number. At some point all platforms are forced to clear their imbalances else/when their membership stops growing.

[12] Despite being named in honor of Keynes's famous international multilateral clearing proposal of 1941, which failed to be implemented at Bretton Woods, the Bancor protocol and its Bonding curve design could only be described as a distant cousin. However, it did have a multilateral clearing currency that was separate from each community currency. Imbalances, rather than being remedied by Keynes' 'fixed but adjustable exchange rates', had programmable continuous exchange rates via a bonding curve that would adjust to balance trade deficits and surpluses (for details regarding the bonding curve see Hertzog et al. 2018).

[13] Like gold or bitcoin, the creation (or mining) of crypto currencies have equity rather than debt as their balance sheet counterpart.

[14] This paper for the most part is focused on transactions between members and chamas, and excludes data analysis of financial transactions or the impact of exchange rate fluctuations in the 2019 period, leaving this to future research. More specifics on the entire ecosystem can be found at https://grassrootseconomics.org/blog (accessed on 27 July 2021).

[15] See Ruddick (2020a) CIC white paper https://gitlab.com/grassrootseconomics/cic-docs/-/blob/master/CIC-White-Paper.pdf (accessed on 27 July 2021).

[16] All the data analysis in Section 4 combines time series for both the 2018/2019 POA blockchain data and the 2020/21 xDai blockchain data. We have excluded all: transfers that are disbursements and conversions or reclamations; business-types that are system, administration, or enrolment agents; smart contract transactions during the POA period; and members with missing region-types presumed to be one of the prior entities. Data included are 'standard' transactions that are assumed to be in exchange for *real* goods or services by the CC members ('beneficiaries') and chamas ('group accounts'). The only caveat is that individual saving deposits and withdrawals with chamas which are not in exchange for goods or services, have not been removed even though they would be considered an internal financial transfer. The full dataset (Ruddick 2021) and its description are available at https://reshare.ukdataservice.ac.uk/855142/ (accessed on 27 July 2021).

[17] https://www.grassrootseconomics.org/post/crisis-supply-chains (accessed on 27 July 2021).

[18] https://data.worldbank.org/indicator/NY.GDP.MKTP.KD.ZG?locations=KE (accessed on 27 July 2021).

[19] Chamas on average are around 30 people and they can open a single CC group account to save and lend Sarafu with each other or use the account for communal business For a description of chamas and their participation with GE see Barinaga (2020).

[20] Several new blockchains have been piloted by Grassroots Economics, but due to erratic fees (gas) a future move is planned to the bloxberg.org blockchain. Bloxberg is an Ethereum Virtual Machine—Proof of Authority blockchain developed and run by universities and research organizations like the Max Planck Digital Library.

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
