# Peer review of "Complementary Currencies for Humanitarian Aid"

_jrfm, doi:10.3390/jrfm14110557_

Round 1

Reviewer 1 Report

(please see pdf attached)

Author Response

We have revised our paper substantially due to the reviewers request for us to be more transparent about a major inconsistency. 

Reviewer 2 Report

I don't have aby major issues. To papier is quite clearly written.

Minor issues:

The are no labels in Fig. 3 and 4

Fig 6 should be bigger.

2x Fig 7. The 2nd one is missing labels.

Author Response

We have fixed these minor issues.

Reviewer 3 Report

The paper deals with a rather unusual topic or, more precisely, combines complementary currencies (which economic literature has often analysed) with humanitarian aid in poor regions of the world. However, the authors reach the goal of contributing with a concrete economic proposal to the state of the art. Therefore, I have two minor suggestions:
1) reducing the amount of words by selectively cutting adverbs, adjectives etc. In fact, the paper is not an easy read (or, at least, doew not precisely fit into the usual economic paper typology) despite having been beautifully written;
2) providing some more general macroeconomic data on the Kenyan use of payment methods. This is, in my opinion, in line with suggestion 1), with the relevance of better describing the payment environment Kenyan people are faced with and contextualizes why complementary currencies might be a useful addition to other forms of humanitarian aid in selected countries.

Author Response

We have revised the paper substantially.

In section 4 we have added this paragraph

Grassroots Economics (GE) and their CC ecosystem originated with William O. Ruddick, a United States Peace Corps volunteer and physicist, who has implemented Kenyan CCs for humanitarian goals since 2010.  GE is focused on people in the lowest wealth quintile in Kenya. The Kenyan Centra Bank 2019 household survey estimated that in this wealth bracket 22% were excluded from accessing financial services which mean they kept money in a secret hiding place and borrowed from family and friends. Add another 15% who had access to informal services like savings groups (chamas), shopkeeper credit, money lenders or employers. Amongst the entire population it was estimated that mobile phones were owned by 91% of people and 79% used them for making payments. Despite the meteoric rise in mobile phone payments with the rise of M-Pesa, cash is still the primary tool for daily expenses. That said, 71% of all mobile phone users had used them to pay an amount that was less than 2.50 KSh in the month. In 2019 2/3 of households would have incur a problem during their income cycle of not being able to meet their daily expenses, and 30% of the time they would go to a chama for help.

Round 2

Reviewer 1 Report

Please see report attached
